# Mechanical Stimulation Decreases Auxin and Gibberellic Acid Synthesis but Does Not Affect Auxin Transport in Axillary Buds; It Also Stimulates Peroxidase Activity in *Petunia* × *atkinsiana*

**DOI:** 10.3390/molecules28062714

**Published:** 2023-03-17

**Authors:** Agata Jędrzejuk, Natalia Kuźma, Arkadiusz Orłowski, Robert Budzyński, Christian Gehl, Margrethe Serek

**Affiliations:** 1Institute of Horticultural Sciences, Department of Environmental Protection and Dendrology, Warsaw University of Life Sciences, Nowoursynowska 159, 02-787 Warsaw, Poland; 2Institute of Information Technology, Department of Artificial Intelligence, Warsaw University of Life Sciences, Nowoursynowska 159, 02-787 Warsaw, Poland; 3Faculty of Natural Sciences, Institute of Horticulture Production Systems, Floriculture, Leibniz University of Hannover, Herrenhäuser 2, 30167 Hannover, Germany

**Keywords:** thigmomorphogenesis, plant architecture, plant hormone synthesis, cell wall lignification

## Abstract

Thigmomorphogenesis (or mechanical stimulation-MS) is a term created by Jaffe and means plant response to natural stimuli such as the blow of the wind, strong rain, or touch, resulting in a decrease in length and an increase of branching as well as an increase in the activity of axillary buds. MS is very well known in plant morphology, but physiological processes controlling plant growth are not well discovered yet. In the current study, we tried to find an answer to the question if MS truly may affect auxin synthesis or transport in the early stage of plant growth, and which physiological factors may be responsible for growth arrest in petunia. According to the results of current research, we noticed that MS affects plant growth but does not block auxin transport from the apical bud. MS arrests IAA and GA_3_ synthesis in MS-treated plants over the longer term. The main factor responsible for the thickening of cell walls and the same strengthening of vascular tissues and growth arrestment, in this case, is peroxidase (POX) activity, but special attention should be also paid to AGPs as signaling molecules which also are directly involved in growth regulation as well as in cell wall modifications.

## 1. Introduction

Mechanical stimulation-MS is a term created by Jaffe (1973), and means plant response to natural stimuli such as wind blow, strong rain, or touch, resulting in a decrease in length and an increase in branching and in the activity of axillary buds [1]. Except for a few plant species (e.g., *Mimosa pudica* and *Dionaea muscipula*), plants used to have a slow response time to MS [2]. It has already been discovered that the youngest tissues are the most responsible for MS. It is worth noting that differences in cultivar responses to touch are also discussed [3]. Several hypotheses claim that MS causes an increase in radial expansion and decreases plant growth on the cellular level by the arrest of auxin synthesis or transport, decrease of gibberellin synthesis, and increase in peroxidase activity [4,5,6,7,8].

MS is very well known in plant morphology, but physiological processes controlling plant growth are not well described yet. MS inducing a reduction in stem elongation of young plants is the result of increases in the activity of indoleacetic acid (IAA) oxidase and other enzymes such as phenylalanine ammonia-lyase or cinnamyl alcohol dehydrogenase [9,10,11]. It is suggested that mechanical stimulation may reduce auxin synthesis in a shoot apex and young leaves and/or its basipetal transport [12,13,14,15]. According to Jedrzejuk et al. [11], in petunia subjected to mechanical stimulation, auxin synthesis was not arrested, but it was hypothesized that auxin transport may be interrupted [11]. Basipetal auxin transport may be facilitated by auxin influx carriers (AUX1/LAX proteins) and by efflux PIN and PGP protein families [16,17,18]. MS, as moderate stress, also may increase peroxidase activity and decrease endogenous gibberellin content [19,20,21,22,23,24,25,26,27]. On the contrary, peroxidases are responsible for cell wall lignification and suberization, which leads to an arrestment of cell elongation [28,29,30].

In the current study, we try to find an answer to the question if MS truly may affect auxin synthesis or transport in the early stage of plant growth as well as which physiological factors may be responsible for growth arrest in petunia.

## 2. Results

### 2.1. Shoot Growth

The most significant differences in shoot growth of studied plants were visible on T_3_ (30 days after the experiment started) and T_4_ (56 days). The most intensive shoot growth was observed in control plants (5.63 and 5.67 cm, respectively) and the least intensive in plants stroked 160 times per day (3.82 cm in both terms). Shoot growth varied according to the intensity of mechanical stress (Figure 1). On T_5_ (15 days after MS was turned off) the most intense shoot growths were observed in plants stroked 120 and 160 times per day (Figure 1).

### 2.2. IAA Content

IAA content varied in the shoot as well as root apices regardless of timing and intensity of stress (Figure 2a,b). In SAM (shoot apical meristem), the lowest IAA content was observed on T_0_ (start of the experiment). On T_1_ and T_2_ (7 and 14 days after the experiment started), the IAA content varied depending on the intensity of stroking (Figure 2a), but it is worth noticing that IAA content in plants treated with MS was always higher than in the control plants in T_2_ (Figure 2a). On T_3_ (30 days after the experiment started), the highest IAA content was observed in control plants, while the lowest content was evident in plants stroked 160 times per day. On the last day of the experiment (15 days after MS was turned off), the highest IAA content was observed in control plants that were stroked 160 times per day.

In RAM (root apical meristem), IAA content varied in plants regardless of MS length as well as stress intensity (Figure 2a,b). On T_0_, T_1,_ and T_2_, there were no significant differences between IAA content in control and plants subjected to MS. On T_3_, IAA content in plants stroked 80 and 120 times per day was significantly higher than in control plants and those stroked 160 times per day. The highest IAA content in control plants and those stroked 80 and 120 times per day, in SAM and RAM, was observed on T_3_ (Figure 2b).

### 2.3. Immunohistochemistry of Auxin Carriers

To complement the data obtained by spectrophotometric analysis, immunolocalization was carried out in apical meristems of stems (SAM) and roots (RAM) of the studied plant. Due to the most significant differences in auxin content between control and plants subjected to MS on T_3_ (plants subjected to MS 120 and 160 times per day), and no differences in auxin transport between both terms, immunodetection results were presented on T_3_. On T_3_, LAX1, AUX1, and PIN1 labeling was detected in meristematic cells of SAM buds as well as provascular strands of the main stem in control plants as well as in plants subjected to MS (Figure 3A,C,E,G,I,K). The immunolocalization signal was not detected in axillary buds (Figure 3B,D,F,H,J). The immunolocalization signal was very well detected in the main axis above and below dormant buds in all studied plants. In root apical meristems (RAM), auxin detection and intensity were similar regardless of the treatment and labeling and regardless of stress duration and intensity (data not presented). Clear auxin careers signal was coming from the epidermis part and vascular tissues of the root. To conclude, there were no differences in auxin transport in control as well as MS-stimulated plants.

### 2.4. Gibberellic Acid (GA_3_) Content

In roots, GA_3_ content was too low to be measured. In SAM, GA_3_ content varied depending on MS intensity and the duration of the stress (Figure 4a,b). On T_2_, GA_3_ content was the highest in plants subjected to MS 80 times per day. On T_3_, the highest GA_3_ content was observed in control plants and those subjected to MS 80 times per day. The lowest GA_3_ content was observed in plants subjected to stroking 120 and 160 times per day. On T_5_, GA_3_ content increased in plants stroked 120 and 160 times per day compared with T_3_, but it was still significantly lower than in control plants. An interesting fact is that in plants stroked 80 times per day, GA_3_ was the lowest on this term. When taking into consideration the duration of MS stress, the highest GA_3_ content was observed on T_3_ in controls and plants stroked 80 times per day, whereas was observed on T_1_ in plants stroked 120 and 160 times per day.

### 2.5. Peroxidase Activity

In SAM, on T_1_ and T_2_, peroxidase activity was fairly high in plants brushed 120 and 160 times (T_1_) or only 160 times (T_2_) versus peroxidase activity in control plants. On T_3_ peroxidase activity was the lowest in plants stroked 120 and 160 times per day. On T_5_, peroxidase activity was higher in all brushed plants than in controls (Figure 5a). On T_1_ and T_2s_, in RAM, high peroxidase activity was observed in plants brushed 80 and 160 times (T_1_) and in all brushed plants on T_2_. On this term, the highest POD activity was 631.5 U, which was two times higher than in control plants (Figure 5a,b). In SAMs, in plants brushed 120 and 160 times per day, the highest peroxidase activity was observed on T_1_, while in RAMs, in all studied plants, on T_2_—14 days after MS started (Figure 5b).

### 2.6. Histological Determination of Cell Wall Lignification in Vascular Bundles of Stems and Roots

Histological observations of stems and roots of examined plants were presented on T_1_, because of the highest POX activity in SAMs. Control and plants subjected to MS 160 times per day were examined. On T_1_, POX activity was the highest in stems of plants subjected to MS 160 times per day. No tracheary elements of vascular bundles were detected on longitudinal sections of stems in control plants (Figure 6A–D); in contrast to plants subjected to MS 160 times per day, where tracheary elements were clearly visible in vascular bundles (Figure 6E–H). We also checked the differences in collenchyma and sclerenchyma cells in control and MS-subjected plants. We observed clear sclerenchyma cells in treated plants, whereas this was not the case in the control plants (Figure 6I,J). There was no difference in root anatomy regardless of plant treatment (data not presented).

## 3. Discussion

Plants may be sensitive to environmental factors such as strong wind and rain, ranging from very rapid and intense to more moderate and slow [31,32,33,34,35,36]. Some responses of plants to mechanical stimuli are very rapid due to the presence of specialized cells [33]. Plants without specialized sensory cells react slowly by changing morphology. According to Biddington (1986), the most common feature of MS is a decrease in the elongation of shoot growth and an increase in radial expansion [37]. These changes enable plants to withstand mechanical stresses [38]. In the current study, petunias subjected to regular mechanical stimulation 30 days after stroking started were 23 to 32% smaller. A previous study on petunia also demonstrated a higher number of branching in two petunia cultivars exposed to MS [11]. The phenomenon of an increase in the diameter of plants subjected to MS was confirmed by the series of time-lapse photography series (Appendix A).

The main factors taking part during the MS process are auxins, gibberellins, cytokinins, peroxidases, and oxidases [39,40,41,42,43,44]. According to former research, mechanical stimulation of soybean and pea plants reversed auxin-promoted shoot elongation [45,46]. Hofinger et al. (1979) demonstrated that auxin, under natural conditions present in the lower internodes of the wild cucumber, was absent after the MS of the plants [47]. In the current research, on T_2_ (14 DAS-days after MS), IAA content in SAM, in all plants subjected to MS, was higher than in unbrushed plants from 72 to 127.8%, while on T_3_ (30 DAS), IAA content was lower in all brushed plants versus controls. The difference in IAA content varied between 200 and 292 µg IAA·g^−1^ DW. The current experiment clearly demonstrated that MS affected auxin synthesis in the SAM of studied plants as late as 30 days after MS started. There is no data confronting the difference in IAA content between SAM and RAM. In the current study, much lower IAA content was observed in RAM than in SAM in all studied plants. The auxin content in roots varied from 71 on T_0_ in all collected plants to 199 µg IAA·g^−1^ DW on T_3_ from plants stroked 80 times per day. 

The main purpose of the current study was to directly answer the question of whether MS arrests the auxin synthesis or transport in petunia, affecting the same plant architecture. In this case, we decided to conduct a series of immunolocalization studies. The most probable route of auxin transport to the RAM is the developing vascular tissue, which is high auxin content reach. It is now well established that auxin transport is facilitated by auxin influx and efflux carriers [48,49,50,51,52,53]. In the current study, we made a series of observations on auxin transport through the LAX, AUX, and PIN protein groups. The observations of active auxin transport were made on the specimen collected in the same terms as for quantitative auxin measurements. At 30 days after the start of MS, we noticed a much lower auxin content in SAM in all treated plants than in control plants, but we did not observe a similar event in roots. 

In plants with strong apical dominance, the shoot apex inhibits the activity of axillary buds. Removal of the shoot apex initiates the activity of axillary buds. This activity is accompanied by PIN carrier polarization, enabling auxin export from the auxiliary buds [14]. Research conducted by Balla (2016) on pea clearly presented that auxin export from axillary buds are only possible if the primary source of auxin is removed or weakened. 

In the current study, on all terms of observations, LAX1, AUX1, and PIN1 labeling were detected in meristematic cells of SAM buds as well as provascular strands of the main stem in all studied plants, while the immunolocalization signal was not detected in axillary buds. The immunolocalization signal was very well detected in the main axis above and below dormant buds in all studied plants, but it was not detectable in stems below dormant buds. This may be the clue to discuss if MS was enough strong to decrease auxin synthesis but too weak to interrupt its basipetal transport. Or whether other physiological factors may be responsible for auxin transport and prevention of axillary bud release. According to Mason et al. (2014), the dogma of auxin-mediated apical dominance has persisted largely that auxin is typically capable of inhibiting the later stages of bud outgrowth after decapitation, and because it regulates the levels of other hormones known to affect shoot branching [54]. However, by observing the earliest stages of bud release, it appeared that auxin depletion was not sufficient to induce bud release after decapitation. The factor responsible for axillary bud release from apical dominance was the level of endogenous sugars. A non-sufficient level of soluble sugars may be a central point to the maintenance of apical dominance. These hypotheses deny the results of the current experiment presenting an increase in plant diameter (see Appendix A). We did not study cytokinin content in petunia SAMs; but according to Qiu et al. (2019), high cytokinin content in apical buds may trigger the burst of non-dormant bud [55].

IAA and GA are strong growth regulators of aerial organs, and the endogenous levels of these hormones quantitatively regulate the shoot growth as accelerators, i.e., the higher the endogenous level the greater the shoot growth [56]. Physiological studies revealed that GA plays an important role in internode elongation [12,15,27,57,58,59]. There is still a lack of understanding of the signal transduction pathways that lead to GA activity and thus to the elongation of stems and leaves in response to different environmental factors. According to Tanimoto (2007), auxin and gibberellins are the strongest accelerators of shoot growth [60]. It has been suggested that auxin transport from the shoot controls root growth by facilitating the GA-mediated destabilization of DELLA proteins [61,62,63,64,65,66,67]. In the current study, we noticed that T_3_ was the critical date to observe clear growth arrestment in plants subjected to MS as well as IAA and GA_3_ content in plants stroked 120 and 160 times per day. There is still a question about the active transport of auxins from the shoot main bud to the roots. According to the literature, decapitation produces a spectacular effect in which the transport of auxins from the main shoot is rapidly inhibited and their transport from lateral shoots is activated. In the current experiment, the effect of reduced synthesis of auxins and gibberellins occurred after 30 days of stress. The main question we wanted to answer in this study was whether MS may arrest auxin synthesis or transport at an early stage of stress. The present experiment yielded a clear answer: under MS, auxin and gibberellin synthesis is inhibited during prolonged stress, while basipetal auxin transport is not inhibited.

Peroxidase-mediated oxidative decarboxylation is crucial for auxin arrest in plant stems [68,69]. An increase in peroxidase activity that occurred in mechanically perturbed plants was noticed by Hofinger et al. (1980) and Boyer et al. (1980) [47,70]. The results of the research made in 2020 on petunia also showed higher activity of peroxidase in plants subjected to MS reverse to controls. In the current study, peroxidase activity was variable in SAM, while in RAM on T_1_ and T_2_, high peroxidase activity was observed in plants brushed 80 and 160 times (T_1_) and in all plants on T_2_. Histological observations of stems and roots of examined plants were presented on T_1_ because of the highest POX activity in SAMs. On T_2_, in stems of plants subjected to MS 160 times per day, peroxidase activity was the highest. An interesting fact is that in SAMs the highest peroxidase activity was observed on T_1_, but no tracheal elements of the vascular bundles were visible on longitudinal sections of the stems in the control plants, opposite to plants subjected to MS 160 times per day, where tracheary elements were clearly visible in vascular bundles and sclerenchyma cells. It is obvious that plants create a natural defense against mechanical damage through the production of lignins and suberins in stems [30]. In tomato plants, peroxidase activities significantly increased in the rubbed internode after mechanical stress application [10]. The results of the current study are consistent with previous studies on the role of peroxidases in the MS response of B. dioica and B. pilosa as well as in tomato plants [10,71]. In the current study, an increase in POX levels in roots resulted in the development of tracheary elements in petunia shoots. It is quite new that the most intense POX activity was observed in roots, and there is not much data about chemical signals appearing in roots caused by MS that have their effects on the stem. According to Potocka et al. (2018), surprisingly, mechanical stimulation seems to have greater effects on root growth than on shoot growth, which has been mainly studied [1,72,73,74].

Results of the previous studies suggest that roots may have a larger response to MS than shoots. One possibility is that the response to mechanical stimulation is a whole plant response, but that the roots have a higher sensitivity to mechanical stimulation [72]. Until now, research comparing root and shoot response to MS focused mainly on morphological, anatomical, and cytological changes in roots or molecular factors activating cytological changes, but there are not much data concerning physiological root response [72,75,76,77]. In the current study, we did not examine how stress caused by MS affects ROS as well as antioxidant enzymatic activity production in stems and roots of stimulated plants. One of the hypotheses explaining such a large POD activity in roots of stimulated plants in the early stage of MS duration may be its production to prevent the plant against oxidative stress caused by MS. This hypothesis may be true in this case, especially since we did not observe any differences in root anatomy in either control or MS-subjected plants. On the other side, the results of Jacobsen et al. (2021) on Arabidopsis showed that the signaling genes linked to ethylene and auxin differential gradients, and transcriptional activation of ROS, were all part of the early response of Arabidopsis roots to MS, which clearly fits our hypothesis [78].

Basipetal auxin transport may be controlled by different types of proteins and peptides in xylem sap [79]. Especially noteworthy are arabinogalactan proteins (AGPs), which are probably active as signaling molecules during MS [80,81]. They are involved in the regulation of plant development and affect cell wall properties [82,83,84,85,86,87,88,89,90]. In roots, AGPs regulate their elongation and differentiation and may regulate the deposition of cellulose microfibrils in shoots [91,92,93,94]. This may be a clue that AGPs, besides POX activity, also play an important role in cell wall modifications of MS-subjected plants. In the current study, we did not study the activity of AGPs in SAM or RAM of petunia. In any case, there are no data in the literature on whether MS may increase AGPs production and affect cell wall modifications in plants.

## 4. Conclusions

In the current research, we confirmed, that over a longer period of time, mechanical stimulation arrests growth dynamics and auxin as well as gibberellins synthesis in petunia.Mechanical stimulation does not arrest basipetal auxin transport.In the current research, we proved that one of the factors affecting the growth of petunia may be peroxidase activity, which is responsible for cell wall lignification and suberization in stems.In the current research, we proved that petunia plants subjected to mechanical stress 160 times a day clearly reduced their growth while they increased their diameter, which is an asset for the production of bedding plants such as petunia.The increase in growth dynamics in petunias after the cessation of mechanical stress is a clear physiological response of plants to return to a state of homeostasis.Besides POX activity, AGPs may also play an important role in cell wall modifications of MS-subjected plants.In the current study, we did not study the activity of AGPs in the SAM or RAM of petunia. In any case, there are no data in the literature on whether MS may increase AGPs production and has the same effect on cell wall modifications in plants, so it could be the next step to understand the physiological reaction of plants to MS.

## 5. Material and Methods

The experiments were conducted in 2019–2022. The plant material, *Petunia* × *atkinsiana*, ‘Pegasus^®^ Special Burgundy Bicolor’ with purple–white rays on the petals, was obtained from the Volmary Polska Company, central Poland, Mazovia) as 4-week-old plantlets at the beginning of February. The plants were placed in a greenhouse at the Warsaw University of Life Sciences, Poland.

### 5.1. Experimental Design

MS of plants started on 22 February and stopped on 20 April. The whole experiment started on 22 February and finished on 5 May. Between 20 April and 5 May, plants were growing without MS to observe the dynamics of their growth without the stressor. MS was provided by the brushing apparatus described in detail by Jedrzejuk et al. (2020) [11], and constructed by University workers. Only shoot apices were subjected to mechanical stimulation, but both shoot and root tips were subjected to biochemical analysis. This decision was dictated by the fact that roots may respond to stress faster and stronger than shoot apices (according to Potocka et al. (2018)).

### 5.2. Biometric Measurements and Biochemical Analyses

There were 90 plants in each treatment. The average increase of studied plants was measured between 22 February and 5 May in 5 terms: T_0_—experiment beginning, T_1_—7 days after MS started, T_2_—14 days after experiment started, T_3_—30 days after experiment started T_4_—56 days after experiment started (end of MS process), and T_5_—71 days after experiment started (15 days after MS was turned off, to check plant behavior). For biometric measurements, plants were divided into three blocks of 15 plants each. The same plants were examined in each term. Between T_2_ and T_3_, for 10 days the growth habit of control and plants subjected to MS 160 times per day was examined by using the time-lapse photography method (Appendix A).

For biochemical analyses, 1.5 cm stem apical meristem (SAM) or root fragments (RAM) were collected from nine plants divided into three blocks of three plants in each treatment and each term (total 180 plants) (three apices per plant per each replication) on the dates as given above, excluding term 0 for roots (too low amount of plant material) and term 4 (56 days after experiment started in both). The exclusion of material collection on term 4, was dictated by plants being too large and flowering, which would have disturbed the analyses. The specimen for all analyses were collected immediately after brushing was stopped.

All analyses were made in nine replicates per treatment. The dry weight was determined by drying three 1-g samples at 105 °C until the weight was constant.

### 5.3. IAA Content

Free IAA was extracted with 80% methanol and assayed spectrophotometrically with the Salkowski reagent [82] at 520 nm (AOE Instruments UV-1600, Taiwan) and expressed as μg IAA per g DW [82].

### 5.4. GA_3_ Content

Gibberellic acid was determined according to the method of Graham and Thomas (1961) and modified for petunia tissues in the presence of absorbance measured spectrophotometrically at 430 nm according to the standard curve for GA_3_ [83]. Total gibberellins content was expressed in ng of GA_3_·g^−1^ DW.

### 5.5. Peroxidase (POD) Activity

Peroxidase activity was determined as previously described by Jędrzejuk et al. (2020) [11]. The peroxidase activity was determined as follows: 500 mg of tissue was homogenized in 3 mL of 0.1 M sodium phosphate buffer (pH 7.0) (Merck, Germany), and centrifuged at 20,000 rpm for 15 min at 4 °C. The assay mixture consisted of 3 mL of 0.5 M pyrogallol (Warchem, Poland), 0.1 mL of enzyme extract, and 0.5 mL of 1% H_2_O_2_ (Warchem, Poland). The activity was estimated spectrophotometrically at 430 nm (AOE Instruments UV-1600, Taiwan) and expressed as units (U) per g^−1^ DW.

### 5.6. Immunohistochemistry of Auxin Carriers

Immunolocalization of AUX1, LAX1, and PIN1 proteins, was performed on longitudinal stem segments of the apex (SAM) and axillary buds 1 cm below SAM as well as apical parts of roots (RAM). The anti-Arabidopsis-PIN1, AUX1, and LAX1 antibodies (AGRISERA, Sweden) also recognized the homologous protein in petunia, which is presumed to be functional orthologs based on expression similarity and localization signal to Arabidopsis. The following antibodies and dilutions were used 1:100 and anti-rabbit secondary FITC (excitation wavelength 490 nm) conjugated antibody (1:500, AGRISERA, Sweden). Samples were viewed under an epifluorescence microscope Zeiss AxioScope A. Images were acquired using a Zeiss Axio Cam MRm digital camera with the AxioVision Software (Version 4.8.2).

### 5.7. Safranin and Crystal Violet Staining

Longitudinal sections of SAM and RAM samples were prepared at 10 µm thickness using a rotary microtome. Samples were stained in 0.1% aqueous safranine (safranin O, Merck, Germany) for 30 min and rinsed three times for 10 min each in warm water at 37 °C. Sections were then oven dried and mounted in immersion oil (Merck, Germany) for fluorescence microscopy. Sections stained with safranine were excited at 488 nm.

For anatomical observations of collenchyma and sclerenchyma cells, samples were handled according to the methods of preparation for electron microscopy [84]. Semi-thin (3 µm) sections were stained with 0.1% aqueous solution of crystal violet (Earchem, Poland) and dried at 70 °C.

### 5.8. Statistical Analysis

Data were analyzed with the general linear model program Statgraphics Centurion XIX 2019 (Statgraphics Technologies, Inc., The Plains, VA, USA), ANOVA 1 was used for measurements of shoot growth and biochemical analyses, and means were compared by the LSD or Tukey–Kramer multiple range test at the significance level α = 0.05.

## Figures and Tables

**Figure 1 molecules-28-02714-f001:**
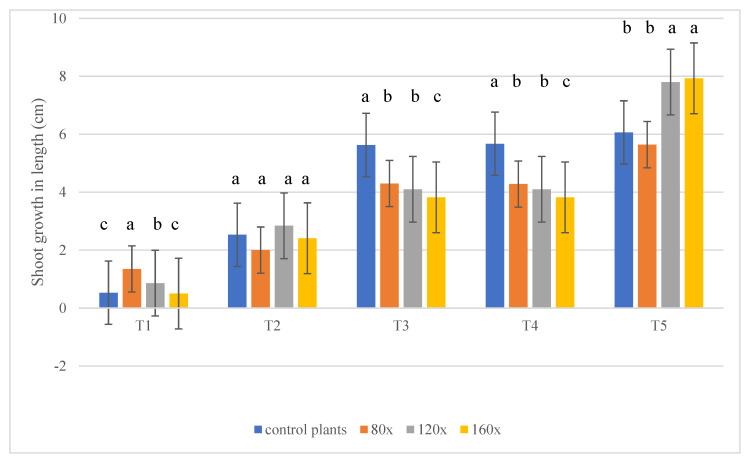
Shoot growth in petunia subjected to MS depending on stress intensity. Statistical analysis was made in each term separately. A total of 15 plants in each block from each treatment were measured in each of the five terms (α − 0.05). The lowercase letters present statistical differences (α − 0.05) between the treatments.

**Figure 2 molecules-28-02714-f002:**
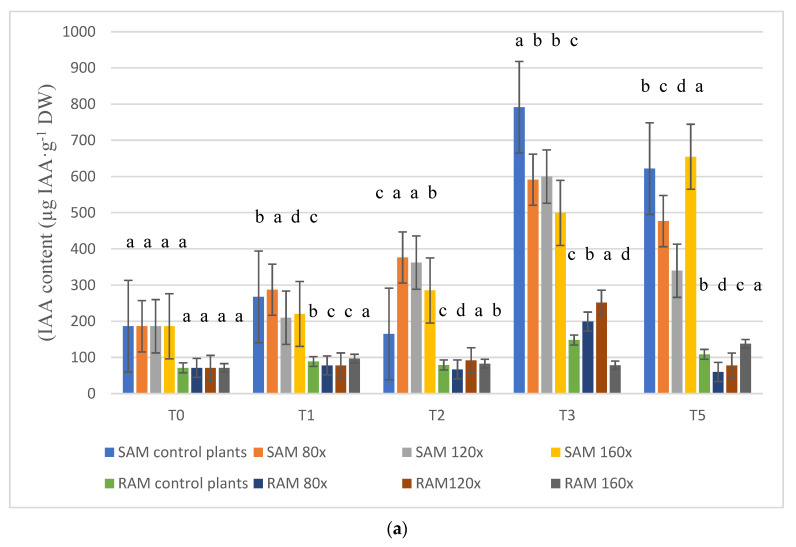
(**a**) IAA content (ng·g^−1^ DW) in the apical meristem of stems [SAM] and roots [RAM], analyzed separately, of petunias subjected to MS depending on stress intensity. The specimen was collected from 15 plants in each block from each treatment in each of the five terms (α − 0.05). (**b**) IAA content (ng·g^−1^ DW) in the apical meristem of stems [SAM] and roots [RAM], analyzed separately, of petunias subjected to MS depending on stress duration. The specimen was collected from 15 plants in each block from each treatment in each of the five terms (α − 0.05). The lowercase letters present statistical differences (α − 0.05) between the treatments.

**Figure 3 molecules-28-02714-f003:**
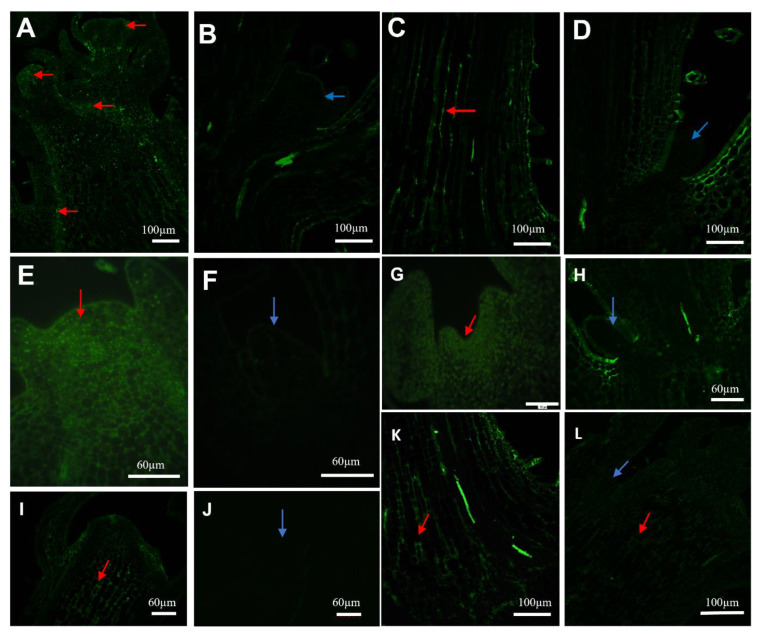
Auxin influx (LAX1 and AUX1) and efflux (PIN1) carriers immunoanalysis in stems of control (**A**,**B**,**E**,**F**,**I**,**J**) and stroked 160 times per day (**C**,**D**,**G**,**H**,**K**,**L**) plants. Red arrows present LAX1, AUX1, and PIN1 antibody signals in SAM buds and provascular strands of the main stem (**A**,**C**,**E**,**G**,**I**,**K**) and no signal (blue arrow) in the axillary bud (**B**,**D**,**F**,**H**,**J**,**L**).

**Figure 4 molecules-28-02714-f004:**
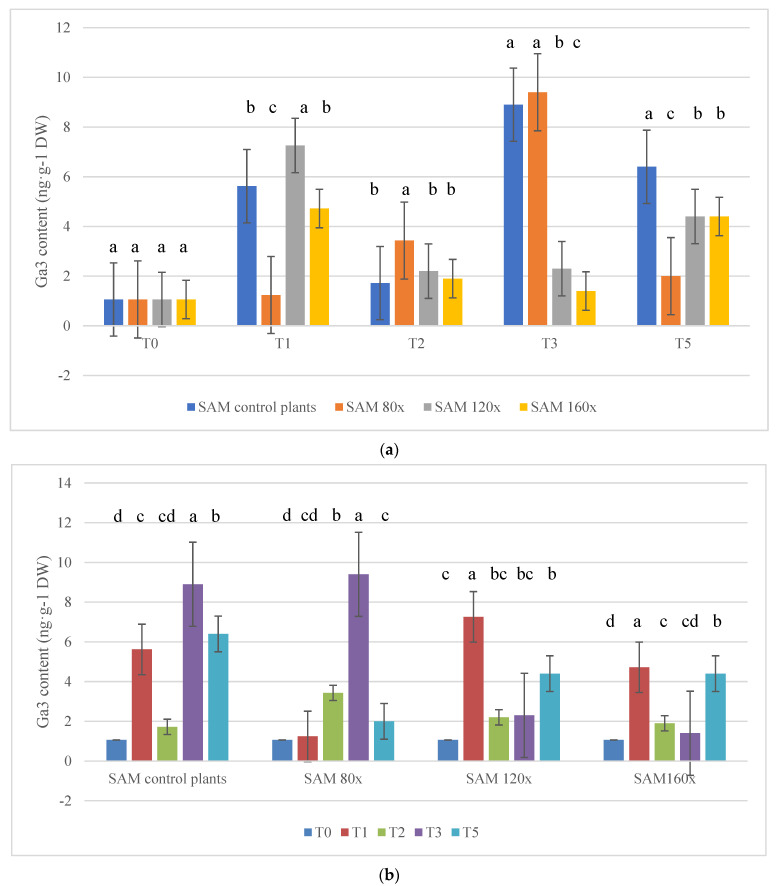
(**a**) Gibberellic acid (GA_3_) content (ng·g^−1^ DW) in the apical meristem of stems [SAM], analyzed separately, of petunias subjected to MS depending on stress intensity. The specimen was collected from 15 plants in each block from each treatment in each of the five terms (α = 0.05). (**b**) Gibberellic acid (GA_3_) content (ng·g^−1^ DW) in the apical meristem of stems [SAM], analyzed separately, of petunias subjected to MS depending on stress duration. The specimen was collected from 15 plants in each block from each treatment in each of the five terms (α − 0.05). The lowercase letters present statistical differences (α − 0.05) between the treatments.

**Figure 5 molecules-28-02714-f005:**
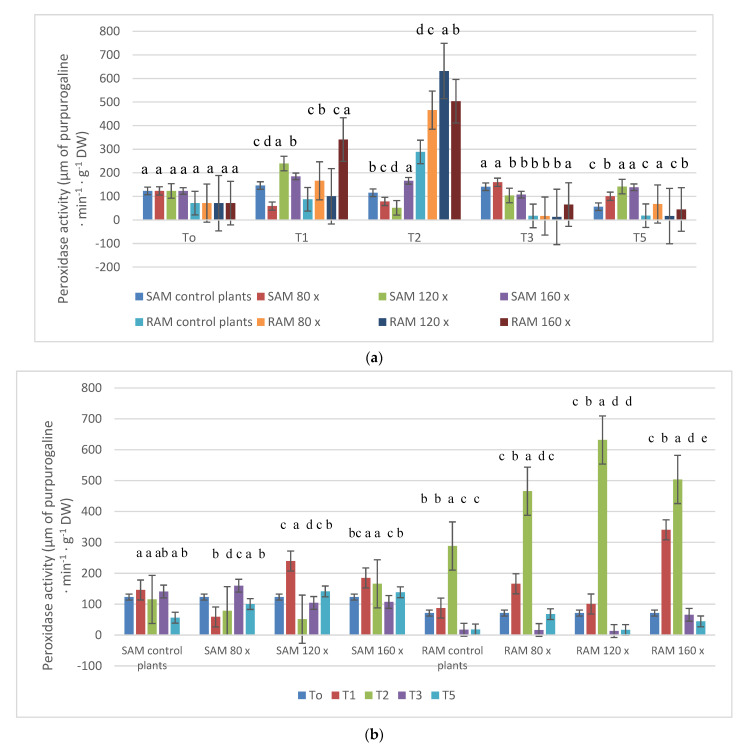
(**a**) POX activity (µg purpurogallin·g^−1^ DW) in the apical meristem of stems [SAM] and roots [RAM] of petunias subjected to MS depending on stress intensity. The specimen was collected from 15 plants from each treatment in each of the five terms; α − 0.05. (**b**) POX activity (µg purpurogallin·g^−1^ DW) in the apical meristem of stems [SAM] and roots [RAM] of petunias subjected to MS depending on stress duration. The specimen was collected from 15 plants from each treatment in each of the five terms; α − 0.05 The lowercase letters present statistical differences (α − 0.05) between the treatments..

**Figure 6 molecules-28-02714-f006:**
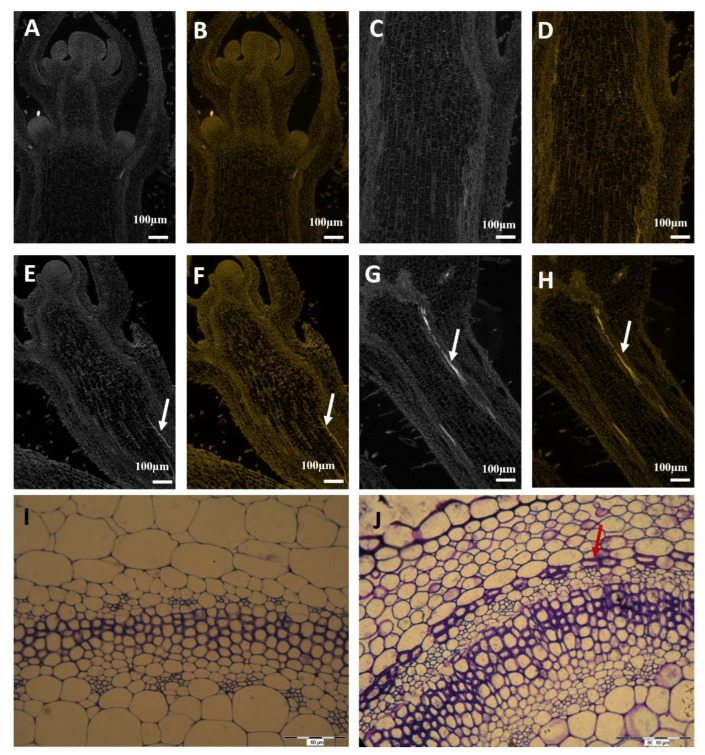
Histological determination of cell wall lignification in vascular bundles of stems of *P. atkinsiana*. (**A**–**D**)—longitudinal section of the stem of control plants. Tracheary elements are not visible in vascular bundles. (**E**–**H**)—longitudinal section of the stem of plants subjected to MS 160 times. Tracheary elements are clearly visible in vascular bundles. Arrow-tracheary elements. (**I**,**J**)—cross-section of the stems of control (**I**) and subjected to MS 160 times (**J**). Usually, plants subjected to stress develop reinforcing tissue, such as collenchyma or sclerenchyma, resulting in the arrest of developing growth. (**I**)—cross-section of the stem of control plants. Sclerenchyma cells are not visible. (**J**)—cross-section of the stem of plants subjected to MS 160 times. Sclerenchyma cells are clearly visible. Arrow—sclerenchyma cells.

## Data Availability

All relevant data in this study are provided in the article and its Appendix A.

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
