# Peer review of "Mechanical Stimulation Decreases Auxin and Gibberellic Acid Synthesis but Does Not Affect Auxin Transport in Axillary Buds; It Also Stimulates Peroxidase Activity in Petunia × atkinsiana"

_molecules, 2023, doi:10.3390/molecules28062714_

Round 1

Reviewer 1 Report

The paper focused on designing experiments to clarify the effect of mechanical stimulation on the auxin synthesis and transport. The overall merit is good. There are some minor points need to be revised.

1. Figure 2a, different columns for RAM should be listed.

2. Figure 6, font size and format should be edited.

Author Response

I would like to thank all the reviewers for the tremendous amount of work put into improving the manuscript. Here is the corrected version of the manuscript. Corrected parts of the manuscript are highlighted yellow.

Figure 2a, different columns for RAM should be listed.

Generally, we created two figures (2a and 2b) to present IAA content in SAM and RAM depending on stress intensity (Fig. 2a) and duration (Fig. 2b). In both Figures columns for RAM are separated from SAM columns. On Fig. 2b created to present IAA content depending on stress duration, columns for SAM and RAM are separated.

2. Figure 6, font size and format should be edited.

Corrected and highlighted yellow.

Reviewer 2 Report

I have the following concerns about this manuscript:

1- In Figure 1, the authors used 15 plants for measurements, I will suggest them to provide Standard error or deviation on the bars. Similarly in all the figures this is missing.

2- Improve the conclusion section and provide solid future recommendations.

Author Response

I would like to thank all the reviewers for the tremendous amount of work put into improving the manuscript. Here is the corrected version of the manuscript. Corrected parts of the manuscript are highlighted green.

 In Figure 1, the authors used 15 plants for measurements, I will suggest them to provide Standard error or deviation on the bars. Similarly in all the figures this is missing.

Standard error bars are added.

2- Improve the conclusion section and provide solid future recommendations.

Corrected according to Reviewer's recommendations (highlighted green).

Reviewer 3 Report

The manuscript submitted is not in the format suggested by MDPI. Line numbering is missing, so it is difficult to indicate the location of suggestions. It is strongly recommended to make double-check throughout the whole document before submitting a manuscript for reviewing. Due to experimental procedures being poorly described, some results don't make sense. e.g. hormone levels quantification in roots and shoots separately. Why? Were both organs

mechanically stimulated? Please, check the punctuation marks throughout the whole document.

Abstract

-Check the format. Thigmomorphogenesis should not be in bold.

- Is it allowed to use references in the Abstract section? Please, check the author's guidelines.

- Suggestion: As indicated in the second line of the abstract, MS covers diverse stimuli, so, please be more specific and indicate what mechanical stimulation was used in this study (mechanical stimulation according to is read in the M&M section)

Introduction

Pag 1. It was already discovered, that… Please, remove that comma.

Pag. 2. It is suggested, that… Please, remove that comma.

            According to Jedrzejuk (2020). Please use the proper format properly to indicate references according to MDPI guidelines. In this case, it should be: According to (number).

Results

Figure format.

-Please, add the error bars for all the graphs of Figure 1, 2, 4, and 5.

-It is not common to split a figure (Fig. 2, 4 and 5). Please, check if it is allowed in the MDPI author guidelines.

Pag 2.

2.1 Shoot increment? Some suggestions: Mechanical stimulation increases/affects? petunia shoot length. Mechanical stress increases/affects? petunia shoot length. I am a

bit confused. According to your contrasts described in the text section, it is understood that control petunia plants had longer shoots than MS-treated plants, but at the end of the experiment (T5) all the MS dosis performances were higher than the control treatment. This observation is related to the comment made in M&M 5.2.

2.2 IAA content. Pag 3. Figure 2. I don't understand this figure. Same captions for panels a and b, but different data. Figure 2a: Something is missing. There are 8 bars by time (T0-T5), but there are only 4 legends indicated (SAM control, 80X, 120X, 160X).

2.3 2.3 Immunohistochemistry of auxin carriers

…labelling (data not presented) Clear auxin careers… A period is missing between presented) and Clear.

2.4 Gibberellic acid (GA3) content

- Figure 4, pag 6.  Same case that Figure 2.

Figure 5, pag 7. Legend for RAM is wrong for the control treatment.

Figure 6, pag 8. More information should be added to the figure caption. Three types of images can be visualized but they are not indicated in the figure caption. Uniforme the font used for the figure caption; the font size is different for the last lines.

M&M

5.1 Experimental design. The MS procedures used in this study  are poorly described. Was the MS a mechanical stimulation? Was only shoot stimulated? Or both shoot or root?  Please, be more specific.

5.2Biometric measurements… Were the lengths of T1-T5 normalized with respect to T0? How was the initial intrinsic variation in the shoot length parameter considered?

Discussion

Pag. 9 …most common features of MS, is a decrease in elongation. Remove that comma.

Pag. 9 The main factors taking part during the MS process, are auxins. Remove that comma.

Pag. The most probable routes of auxin transport to the RAM, is. Remove that comma.

There is an IAA peak around T3, which is observed in all treatments evaluated. Any idea about it? Same case for GA3.

There is a discordance between quantitative IAA analysis and IAA immunolocalization. e.g. In T3,Fig. 2B, shoot, there is a statistically significant difference control and 160x . Transitory IAA changes? Any idea? In the discussion, the authors just rephrase the results about auxins localization, but there is no genuine discussion about this discordance between the quantitative analyses and histochemistry.

Author Response

I would like to thank all the reviewers for the tremendous amount of work put into improving the manuscript. Here is the corrected version of the manuscript. Corrected parts of the manuscript are highlighted pink.

The manuscript submitted is not in the format suggested by MDPI. Line numbering is missing, so it is difficult to indicate the location of suggestions. It is strongly recommended to make double-check throughout the whole document before submitting a manuscript for reviewing. Due to experimental procedures being poorly described, some results don't make sense. e.g. hormone levels quantification in roots and shoots separately. Why? Were both organs

The manuscript was corrected according to MDPI format.

Abstract

-Check the format. Thigmomorphogenesis should not be in bold.

- Is it allowed to use references in the Abstract section? Please, check the author's guidelines.

- Suggestion: As indicated in the second line of the abstract, MS covers diverse stimuli, so, please be more specific and indicate what mechanical stimulation was used in this study (mechanical stimulation according to is read in the M&M section)

Abstract was corrected according to Reviewiewer's reccomendations. The term mechanical stimulation was explained in M&M section.

Introduction

Pag 1. It was already discovered, that… Please, remove that comma.

Pag. 2. It is suggested, that… Please, remove that comma.

            According to Jedrzejuk (2020). Please use the proper format properly to indicate references according to MDPI guidelines. In this case, it should be: According to (number).

Corrected

Results

Figure format.

-Please, add the error bars for all the graphs of Figure 1, 2, 4, and 5.

-It is not common to split a figure (Fig. 2, 4 and 5). Please, check if it is allowed in the MDPI author guidelines.

Standard error bars were added in all Figures. Split of the Figures was checked in MDPI author's guidline.

Pag 2.

2.1 Shoot increment? Some suggestions: Mechanical stimulation increases/affects? petunia shoot length. Mechanical stress increases/affects? petunia shoot length. I am a bit confused. According to your contrasts described in the text section, it is understood that control petunia plants had longer shoots than MS-treated plants, but at the end of the experiment (T5) all the MS dosis performances were higher than the control treatment. This observation is related to the comment made in M&M 5.2.

Shoot increment was corrected onto shoot growth. The idea of the experiment was to observe physiological response of plants mechanically stimulated (stressed) and to check their behavior 2 weeks after stress stopped. It is explained in M&M section.

2.2 IAA content. Pag 3. Figure 2. I don't understand this figure. Same captions for panels a and b, but different data. Figure 2a: Something is missing. There are 8 bars by time (T0-T5), but there are only 4 legends indicated (SAM control, 80X, 120X, 160X).

corrected

2.3 2.3 Immunohistochemistry of auxin carriers

…labelling (data not presented) Clear auxin careers… A period is missing between presented) and Clear.

corrected

2.4 Gibberellic acid (GA3) content

- Figure 4, pag 6.  Same case that Figure 2.

Figure 5, pag 7. Legend for RAM is wrong for the control treatment.

corrected

Figure 6, pag 8. More information should be added to the figure caption. Three types of images can be visualized but they are not indicated in the figure caption. Uniforme the font used for the figure caption; the font size is different for the last lines.

Corrected (higlighted yellow)

M&M

5.1 Experimental design. The MS procedures used in this study  are poorly described. Was the MS a mechanical stimulation? Was only shoot stimulated? Or both shoot or root?  Please, be more specific.

Corrected in the text and highlighted pink (lines 300 - 302)

5.2Biometric measurements… Were the lengths of T1-T5 normalized with respect to T0? How was the initial intrinsic variation in the shoot length parameter considered?

Generally, plants were divided on to three blocks of 15 plants. Each block was measured separately, but plants were the same. On T0 plants, the most similar in growth were chosen to the future research.

Discussion

Pag. 9 …most common features of MS, is a decrease in elongation. Remove that comma.

Pag. 9 The main factors taking part during the MS process, are auxins. Remove that comma.

Pag. The most probable routes of auxin transport to the RAM, is. Remove that comma.

Corrected

There is an IAA peak around T3, which is observed in all treatments evaluated. Any idea about it? Same case for GA3.

Explained in lines 228 - 238

There is a discordance between quantitative IAA analysis and IAA immunolocalization. e.g. In T3,Fig. 2B, shoot, there is a statistically significant difference control and 160x . Transitory IAA changes? Any idea? In the discussion, the authors just rephrase the results about auxins localization, but there is no genuine discussion about this discordance between the quantitative analyses and histochemistry.

Explained in lines 228 - 238

Round 2

Reviewer 3 Report

In the current version of the manuscript, the authors have addressed most of the comments about the figure captions and manuscript format. Although, the changes made to improve the discussion section were minimum (only one paragraph 228-237). The discussion was a weak part of the manuscript since the original version submitted. The authors bring to the discussion some data from previous work but just recapitulate the results (e.g., paragraphs in 190-199, 200-212, 242-251); there is no intent to confront, explain or compare their results obtained in this study with previous reports. Importantly,  some contradictions in this study (e.g. auxins quantification vs auxins immunolocalizations) are not included in the discussion. 

Minor observations.

A critical comparison in this study was the mechanical stress effects between shoots and roots; however, there are no bases to support it. As a reply to this observation made in the previous reviewing process, the authors have included two sentences in the M&M section (300-302) but there are no references. Please add the references.

As Molecules is an electronic journal with no limits to pages, there is no reason not to describe protocols used in the study. Please, describe the protocol used for POD activity (332-334).

Author Response

Answers to the Reviewer:

We would like to express our appreciation to the Reviewer for his/her helpful advice. We have made some changes to the Discussion section to provide additional information on auxin content and transport, including immunolocalization slides, to compare Balla's results with our own. We have also discussed the sugar content and its possible role in axillary buds burst.

Furthermore, we have included additional literature data on the sensitivity of roots to MS compared to shoots and discussed the reasons for observing high POD activity in roots.

We would like to clarify that this manuscript is a continuation of research provided by Jędrzejuk et al. (2020), and we have not repeated the data presented in the former article in the Discussion. We believe that the Discussion is now more appropriate for publication in Molecules.

We have also added information on the methodology of POD activity and updated the literature list. All changes made in the manuscript have been highlighted in yellow.

English was corrected.